# Challenges in the Application of Circular Economy Models to Agricultural By-Products: Pesticides in Spain as a Case Study

**DOI:** 10.3390/foods12163054

**Published:** 2023-08-15

**Authors:** Paz Otero, Javier Echave, Franklin Chamorro, Anton Soria-Lopez, Lucia Cassani, Jesus Simal-Gandara, Miguel A. Prieto, Maria Fraga-Corral

**Affiliations:** 1Nutrition and Bromatology Group, Department of Analytical Chemistry and Food Science, Faculty of Science, Universidade de Vigo, 32004 Ourense, Spain; paz.otero@uvigo.es (P.O.); javier.echave@uvigo.es (J.E.); franklin.noel.chamorro@uvigo.es (F.C.); anton.soria@uvigo.es (A.S.-L.); luciavictoria.cassani@uvigo.es (L.C.); jsimal@uvigo.es (J.S.-G.); mprieto@uvigo.es (M.A.P.); 2Instituto de Investigaciones en Ciencia y Tecnología de Materiales, Consejo Nacional de Investigaciones Científicas y Técnicas (INTEMA-CONICET), Av. Colón 10850, Mar del Plata 7600, Argentina

**Keywords:** circular economy, bioeconomy, agri-food by-products, bioactive compounds, pesticides, food safety

## Abstract

The income and residue production from agriculture has a strong impact in Spain. A circular economy and a bioeconomy are two alternative sustainable models that include the revalorization of agri-food by-products to recover healthy biomolecules. However, most crops are conventional, implying the use of pesticides. Hence, the reutilization of agri-food by-products may involve the accumulation of pesticides. Even though the waste-to-bioproducts trend has been widely studied, the potential accumulation of pesticides during by-product revalorization has been scarcely assessed. Therefore, in this study, the most common pesticides found in eight highly productive crops in Spain are evaluated according to the available published data, mainly from EFSA reports. Among these, oranges, berries and peppers showed an increasing tendency regarding pesticide exceedances. In addition, the adverse effects of pesticides on human and animal health and the environment were considered. Finally, a safety assessment was developed to understand if the reutilization of citrus peels to recover ascorbic acid (AA) would represent a risk to human health. The results obtained seem to indicate the safety of this by-product to recover AA concentrations to avoid scurvy (45 mg/day) and improve health (200 mg/day). Therefore, this work evaluates the potential risk of pesticide exposure through the revalorization of agri-food by-products using peels from citruses, one of the major agricultural crops in Spain, as a case study.

## 1. Introduction

The overpopulation reached in recent decades, which is expected to continue to grow to up to 10 billion people by 2050, demands high volumes of food [1]. Nowadays, production systems are already boosting the overuse of natural resources to increase the production volume, which had led to increases in pollution levels and the volume of generated residues [2]. These factors, among others, have been directly associated with climate change [3]. Despite this, current productive systems may not be able to provide enough healthy food for the future global population [1]. Therefore, a shift in the productive systems is urgent to improve their throughput. The establishment of Bio-Districts that implement agroecology, the improvement of aquaculture productivity or the re-valorization of agri-food by-products are some suggested solutions [4,5]. In fact, waste management has become a tough task with economic and environmental repercussions that have awakened international concern and resulted in a rise in alternative productive systems [2,5].

Over the last decade, several strategies adopted by the European Commission have reinforced the implementation of these alternative productive models. The first European strategy, the bioeconomy, was adopted in 2012 to address the production of renewable biological resources and their conversion into vital products and bioenergy [6]. Then, in 2015, the United Nations adopted the Sustainable Development Goals (SDGs), a set of 17 global objectives to improve life for people and the planet, that the European Commission would later adopt in 2019 [7,8]. The SDGs comprise 169 target objectives to address Agenda 2030 global challenges such as poverty, education, economic growth, climate change, innovation, health and peace [8]. At the same time, at the end of 2015, the EU adopted a comprehensive circular economy policy package, the first Circular Economy Action Plan (CEAP). It included 54 actions and four legislative proposals on waste regarding landfill, reuse and recycling [9]. The CEAP was renewed in 2020 and represents one of the building blocks of the European Green Deal adopted at the end of 2019. The aim of the Green Deal is to reduce net greenhouse gas (GHG) emissions by at least 55% by 2030 [10]. As a part of this Green Deal, the Common Agricultural Policy was implemented, which includes the ‘farm to fork’ strategy, among others. Therefore, this ‘farm to fork’ strategy, presented by the European Commission in May 2020, also represents another key action under the European Green Deal frame. This strategy aims to achieve climate neutrality by 2050 by shifting the current EU food system towards more sustainable models [11]. 

A circular economy mainly consists of replacing the ‘end-of-life’ concept with the reduction, alternative reuse, recycling and recovery of materials considered as waste to improve the environmental quality, create economic prosperity and social equity, and thus achieve sustainable development [12]. This transformation and reutilization of waste could ultimately provide added-value products. Tightly related to a circular economy, the term bioeconomy may be introduced, which is mainly focused on the reduction in GHG emissions to counteract climate change. A bioeconomy covers any sector that uses biological resources to produce food, feed, bio-based products, energy and services. The resulting bio-based products are considered to have a lower carbon dioxide footprint when compared to fossil-fuel-based analogues [13]. Therefore, the circular economy and bioeconomy are sustainable models that have gained global attention in recent years in policy, scientific research and business discussions [14]. These models offer common outcomes regarding economic, ecologic and social benefits through different approaches. Even though neither offer the holistic and integrative model required to overcome current environmental issues, they are still recognized as sustainable alternatives [14]. Indeed, waste-to-bioproducts has become a trend in agricultural production [15], since properly managed waste biomass may trigger the development of new value chains characterized as more profitable and sustainable which, ultimately, may ameliorate the survival of natural ecosystems [13].

However, the reutilization of some by-products (peels, pulp and water mill) as raw materials may imply the accumulation of pesticides, especially when obtained from conventional agricultural systems. This work aims to explore the presence of frequently detected pesticides in major crops to assess the risk associated with the reutilization of their by-products as a matrix to obtain ingredients destined for human consumption.

### Theoretical Background and Literature Review

Regarding global productive industrial sectors, agriculture stands out, with remarkable data on the production of edible and waste biomass. The most abundant worldwide crop is cereals, followed by sugar cane, vegetables, fruits (including citrus fruits), roots and tubers, oil crops and legumes (Figure 1) [16]. Of the total geographical area of Europe (2342 × 10^6^ ha), around 20% (466 × 10^6^ ha) was used for agriculture purposes in 2022, representing almost 10% of the global agricultural lands (4772 × 10^6^ ha) [17]. In terms of production, among the EU countries, France and Germany stand out for cereals, roots and tubers and legumes, while Spain and Italy lead the contribution of fruits, citrus fruits, vegetables and olives. These high production rates are associated with the generation of huge volumes of waste, mainly due to maintenance and/or processing steps. The last JRC report that quantitatively assessed biomass production in the EU indicated that between 2016 and 2020, the average was 924 Mt per year. Primary products (mainly edible products) represent 54%, whereas the remaining 46% are secondary products (leaves and stems) [18]. Six EU countries (France, Germany, Poland, Spain, Italy and Romania) are responsible for 70% of the economic (384 Mt/year) and residue production (295 Mt/year). Regarding residue production, cereals are the major contributor, followed by oil-bearing and permanent crops, especially in Spain, Italy, Greece and Portugal [18]. 

In 2022, Spain, apart from being the top European producer of vegetables, fruits and olives, was the second largest producer of legumes and the third largest producer of cereals, sugar cane and roots and tubers (Figure 1) [16]. The highest production yields are usually obtained for the cultivation of olives; fruits like citrus, grapes or berries; culinary vegetables like potatoes, tomatoes or peppers; and different types of cereals like wheat or rice, among others. However, as previously pointed out, apart from creating a great volume and variety of food and feed, a parallel huge volume of waste is also generated (Table 1). Following the EU and global legal context, Spain presented the first “Strategy of Bioeconomy: Horizon 2030” in 2018 with application to all economic activities, including agriculture [19]. A recently published law aims to reduce waste production and to regulate waste treatment based on a principle of waste hierarchy (Figure 1) and considering a circular economy [20]. One of the main objectives of the waste hierarchy is to provide food ingredients for human consumption. Thus, the reutilization of agricultural waste to recover edible biomolecules seems a feasible approach to comply with the current sustainability challenges. Moreover, a broad variety of Spanish horticultural production is also present in biomass waste [21,22]. Hence, agro-industrial residues possess a great potential to recover a wealth of bioactive compounds, especially from by-products like pulp or peel derived from processing steps. Other abundant biomasses derived from crop maintenance, such as leaves or stalks, are also considered great sources of biomolecules [23]. Pigments, flavonoids, tannins, phenolic acids, unsaturated fatty acids and vitamins are some of the most precious bio-compounds present in agricultural biowaste [24]. In the last decade, these molecules have attracted attention because of their associated beneficial bioactivities, such as antioxidant or antimicrobial [25], but also for their and rheological properties, such as jellifying or emulsifying [26,27]. Therefore, recovered biomolecules from agricultural waste may present a wide range of applications in the food industry, like natural additives, to improve the organoleptic properties of food or even to formulate functional foods [28,29,30].

For instance, olive bagasse or peels are generated in large volumes and represent an excellent raw material to be exploited [41]. In addition, as the olive tree is a perennial, its leaves also constitute a constant source of biomass [42]. Olive bagasse and peels contain hydroxytyrosol, oleorupein, nuezhenide, verbascoside, pectins and xyloglycans [43]. Wastewater from the washing and pressing of olives showed remarkable amounts of fermentable oligosaccharides and phenols, such as catechol, whereas olive leaves contain pigments or alkenes, such as 2-deenal-(E) [42].

Another important productive sector is the cultivation of grapes which provides fresh fruits for direct consumption and also for wine production. Indeed, grape processing for wine production is an important source of residues. Overall, pigmented grapes are very rich in phenolic compounds, especially in anthocyanins (delphinidin, cyanidin, petunidin, peonidin or malvidin), although they may also contain other flavonoids like flavan-3-ols or flavonols, phenolic acids, tocopherols and tannins [44,45]. 

A relevant part of citrus cultivation is aimed at the production of juice, which also generates a significant volume of residues such as discarded leaves, skin and pulp. Citrus peels are very rich in flavonoids, such as naringenin or naringin, and terpenes that are well recognized for their aromatic properties, such as limonene [46]. Indeed, one of the most relevant extracts obtained from citrus fruit residues is essential oils. In addition, citrus fruit can is a matrix rich in carotenoids, such as β-carotene or lutein; they also have a high content of phenolic compounds (ferulic, caffeic, gallic, protocatechuic and 4-hydroxybenzoic acids) and flavonoids (flavan-3-ols such as catechin and epigallocatechin, flavanones like luteolin, apigenin and vitexin and the flavonol rutin) and significant levels of pectin, even when compared to other fruits and vegetables [47,48]. Finally, citrus peel is also rich in ascorbic acid [49,50].

Regarding tomato production, their most abundant residues comprise peels and seeds, which are derived from processing them to obtain sauce. The pigment composition of tomato peels has been deeply studied; it consists of carotenoids such as lutein, β-carotene and lycopene, the latter being the major representative [51]. In addition, both tomato seeds and peels also contain high levels of quercetin-3-rutinoside and phenolic acids such as caffeic and chlorogenic acids [52]. 

In the case of red berries like strawberries, Spanish production represents an important contribution to the European market. Discarded strawberry pulp is an important source of phenolic acids, anthocyanins, flavan-3-ols such as catechin and flavonols like rutin [53]. In addition, strawberries present high levels of ascorbic acid; in fact, a portion of strawberries can even exceed the average content in a portion of citrus fruit [54]. 

The main cereal residues include bran resulting from processing steps and stems obtained from harvesting. Even though bran residue has been reduced in the last decade due to the intensified consumption of whole cereals, the cellulosic fraction obtained in the production of refined flours is still an important waste. Cereals like wheat, oats and sorghum are considered a rich source of dietary fiber, phenolic acids (ferulic acid), tocopherols and β-sitosterol [55].

Regarding the specific applications of bioactive compounds, they can be included in food matrixes in different ways. They can be transformed into lyophilized flours and used to recover essential oils or to extract biomolecules. Biomolecules may be recovered as a complex mixture of molecules or as a purified extract. Purified bioactive compounds recovered from residues can be prepared and administered as food supplements. Alternatively, purified biocompounds can be used for the development of nutraceutical foods and as food additives like colorants (pigments such as carotenoids or anthocyanins), preservatives (due to their antioxidant and antimicrobial capacities) or fortifiers (as vitamin fortification), among many others [56]. After agricultural wastes are lyophilized and crushed to produce flour, they may be incorporated into foodstuffs as a part of the final mixture with other flour (meat, fish, fruits, cereals, etc.). This approach provides new organoleptic characteristics that may have a positive impact and thus benefits are gained by the incorporation of bioactive compounds [57]. In the specific case of the utilization of dietary fibers such as celluloses, hemicelluloses or pectin, they can add prebiotic functions [27]. Likewise, depending on the administration route, dietary fibers can serve to develop functional foods’ nutraceutical properties in bakery or pastry products, pickles, dairies, juices and sausages [26]. On the other hand, dietary fibers can also achieve the role of a natural food preservative and act as a substitute for artificial antioxidants such as the currently widely used butylhydroxyanisole, butylhydroxytoluene or propylgalate. In this sense, very different ingredients obtained from natural sources have been widely analyzed as potential food preservatives. Several studies have underlined that pure compounds or extracts such as essential oils are able to extend the shelf-life of fruits, vegetables, meat, fish and canned products. Indeed, some of the tested extracts have been demonstrated as more efficient than their synthetic analogues [58]. In addition, essential oils obtained from agricultural residues can be included as an aroma in food matrixes or as part of active packaging where they can also exert preservative functions. 

This bloom of scientific and industrial research works has been boosted because of the necessity of reducing the volume of agricultural waste. In addition, the current consumption model has undergone a switch that is generating a new market trend. Nowadays, conscious consumers seem to appreciate the presence of natural ingredients as well as unprocessed and environmentally friendly products. Hence, the reutilization of residues to obtain extracts rich in bio-compounds may provide environmental and economic benefits. However, the potential presence of pesticides in some kinds of waste, such as peels, may represent a safety risk since high concentrations of pesticides may be reached when extracting and purifying biomolecules. To the best of our knowledge, this approach has never been considered before. The aim of this work is to investigate the presence of pesticides in eight of the most important agricultural crops in Spain, whose by-products can be targeted for the recovery of biomolecules, to estimate the environmental and human and animal health implications of the revalorization of biowaste. The presence of pesticides will be assessed in potatoes, tomatoes, grapes, oranges, strawberries, peppers, olives and cereals. 

## 2. Methods

### 2.1. Determination of Pesticide Presence in Major Crops in Spain

The scientific literature was examined by using several combinations of keywords as search terms in Google Scholar. The keywords ‘pesticide’ and ‘Spain’ were searched combined with the name of each crop type but also with the general keywords ‘fruit’ or/and ‘vegetable’. For a deeper search, the same approach was followed but replacing the keyword ‘pesticide’ with ‘residue’. From the obtained results, reviews were excluded and only experimental articles where pesticide determination was performed were selected. Additionally, reports carried out on an annual basis by the European Food Safety Authority (EFSA) to determine the presence/absence of pesticides were included. In these reports, the pesticide residue levels in 12 individual food commodities of the European markets are assessed. These same 12 food commodities are targeted every 3 years, which allows the establishment of a trend in terms of pesticide levels. In addition, a paper published in 2022 regarding notifications on pesticide residues in food submitted to the Rapid Alert System for Food and Feed (RASFF) for the period between 2002 and 2020 was also included.

### 2.2. Determination of Adverse Effects of Pesticides

A brief review of pesticides’ effects on human and animal health and the environment was performed in this section. The main aim was to subtly introduce the reader to the wide variability of adverse effects associated with pesticide exposure. Searches were performed in Google Scholar using different terms for each purpose. For instance, for the establishment of acute and chronic exposure of pesticides, the terms ‘organochlorides’, ‘organophosphates’, ‘carbamates’, ‘pyrethroids’ and ‘triazines’ were combined with terms such as ‘hazards’, ‘risk assessment’, ‘human/animal health’ or ‘environment’. However, these searches provided an uncountable number of outcomes, mainly reviews with very close titles and very similar content. Therefore, the selection of journal publications was randomly performed, except for two of the publications belonging to researchers from our institution [59,60]. Nevertheless, to minimize the impact of the semi-systematic review and to provide robust information, publications of internationally recognized organizations such as the WHO, the European Commission or the EFSA were also incorporated in this section. Indeed, of the 20 references compiled in this Section, 6 belong to the EFSA, 1 to the European Commission and 1 to the WHO [61,62,63,64,65,66,67,68].

### 2.3. Safety Assessment of Citrus By-Product Revalorization

A safety assessment of the revalorization of citrus fruit peels to recover ascorbic acid was performed. For the development of this safety assessment, several factors were taken into consideration. The research stages in performing this safety assessment are presented in Figure 2.

#### 2.3.1. Selection of By-Products and Major Biomolecules

The selection of the biomass by-product, citrus peels, was made based on different reasons. On the one hand, oranges were chosen for being one of the fruit categories that most often arose in RASFF notifications on pesticides between 2002 and 2020 [69]. On the other hand, oranges were targeted as food products in three annual EFSA reports (2014, 2017 and 2020), which permits an understanding of the pesticide profile in this food commodity and its temporal evolution. Moreover, Spain is the major producer of citrus fruits in Europe [16]. Finally, citrus fruits are well-known sources of ascorbic acid (AA), so information regarding its concentration was easily found.

#### 2.3.2. Harmonization of AA Results: ‘Water Content’ Factor

The concentration of AA was determined on a fresh weight or dry weight basis, depending on the study. Residues were quantified using fresh weight products. Therefore, the results of AA concentration needed to be standardized. To this aim, the water content of citrus peels was investigated. The water content in citrus fruits was generally estimated as 80% in one study [70], whereas other work presented a range of values of 10–14.2% for dried citrus waste and of 72.5–87.0% for fresh citrus waste [71]. Taking into consideration these two outcomes, an average value was recalculated for dry weight (dw; 12.1%) and for fresh weight (fw; 79.75%). Therefore, to standardize the AA concentration, a ‘water content’ factor of 6.6X was applied.

#### 2.3.3. Biomass Weight Required to Fulfill Recommended Dietary Allowance of AA

The recommended dietary allowance (RDA) for AA is widely variable. It depends on several factors such as the relevant health authority, the physical and health status of each individual person or the final objective of AA administration. Indeed, the RDA for AA is highly variable since different purposes are considered, for example, the lowest limit (45 mg/day) is advised to avoid scurvy and the highest limit (200 mg/day) aims to improve health status [72]. Considering both RDAs, the biomass of citrus fruits necessary to fulfill both limits was determined. Citrus fruit peels with lower AA concentrations require a higher amount of biomass to recover enough AA.

#### 2.3.4. Selection of Pesticides

Oranges were selected as food commodities in the EFSA reports of 2014, 2017 and 2020. The EFSA report from 2017 explains that “as in 2014, imazalil and thiabendazole were the two pesticides mostly used in oranges as post-harvest treatment” [73]. Moreover, in the EFSA report in 2020, imazalil was quantified15 times in citrus fruits, 14 of which exceeded the corresponding maximum residue level (MRL). In the same EFSA report, thiabendazole was reported just twice in citrus fruits, but in both cases, it was quantified above the current MRL [74]. Therefore, these two pesticides were selected to evaluate the potential exposure that may involve the reutilization the reutilization of citrus peels for AA recovery. To provide an extreme scenario of exposure, and since they have been mainly detected over their respective MRLs, it was assumed that imazalil was present at 4 or 5 mg/kg (4 mg/kg for lemons, limes and mandarins or 5 mg/kg for grapefruits and oranges) and thiabendazole at 7 mg/kg in all the citrus peels [75,76].

#### 2.3.5. Application of Processing Factor

Finally, a processing factor was calculated based on the efficiency of washing procedures to remove pesticides. Three works presented different washing methods and variable percentages of pesticide removal. Hence, the processing factor was calculated as the average of the outcomes of three published works that used water-based washing procedures [77,78,79]. The processing factor determined for thiabendazole was 45% and 54% for imazalil.

## 3. Results and Discussion

### 3.1. Pesticide Content in the Most Abundant Agricultural Crops in Spain

In the EU, Commission Regulation (CE) 396/2005 establishes the MRL for pesticides in foods and feeds to guarantee consumer protection [80]. In addition, other CEs establish coordinated multiannual community control programs that cover a period of 3 years, such as the last Commission Implementing Regulation (EU) 2020/585 that covers from 2021 to 2023. These programs ensure compliance with pesticide MRLs to assess consumers’ exposure to these residues when consuming foodstuff with plant or animal origin [81]. Thus, this continuous control of the MRLs of pesticides has become a key tool to maintain consumer protection at the maximal high standards.

Among the current extraction techniques implemented to recover multiple residues simultaneously, the QuEChERS (quick, easy, cheap, effective, rugged and safe) method has become key in multi-residue extraction protocols since it implies a one-step extraction using a buffered acetonitrile and MgSO_4_ for precipitation purposes. Then, a clean-up is performed using solid-phase extraction (SPE), which facilitates the removal of organic acids, excess water and other components. This reliable, simple, rapid and effective recovery method is available from international official standardization bodies such as the European Committee for Standardisation (CEN) [82], the Association of Official Analytical Chemists (AOAC) International [83] or the European Reference Laboratories (EURL) [84]. However, when facing complex matrixes, it may require additional extraction steps to minimize ionic suppression during posterior analysis, which ultimately can mask the presence of pesticides or create artefacts which make the correct interpretation of results difficult. Independent of the applied extraction protocol, the most used methods for identifying pesticides include liquid chromatographic (LC) or gas chromatographic (GC) techniques, commonly coupled to mass spectrometry (MS) or tandem mass spectrometry (MS/MS) [85]. In fact, standardized LC-MS and GC-MS multi-detection methods for pesticides, as well as the associated databases and some validated extractive methods, have been published by several international reference laboratories such as those previously mentioned (CEN [82], AOAC International [83] or the or EURL [84]). Standardization of the extractive and analytical protocols is key for the development of inter- and intra-laboratory comparisons. Apart from providing evidence of the robustness of the analytical methods, they additionally permit to evaluate pesticide distribution in agricultural products among different European countries. In a recent publication, the efficiency of three extractive methods (QuEChERS, ethyl acetate and Dutch mini-Luke) was evaluated for three vegetal matrixes: tomato, orange and avocado. These methods permitted the simultaneous detection of 47 pesticides with recovery rates between 70 and 120% (with a relative standard deviation of ≤20%). Calibration curves showed a concentration range between 0.002 and 0.100 mg/L and an R^2^ of >0.99, while matrix interferences were variable depending on the sample type and the target analyte [86]. Therefore, the optimization of these detection methods is critical to obtain reliable and robust results to identify and quantify analytes accurately.

Regarding the specific case of Spain, several works have underlined the presence of high concentrations of pesticides in Spanish agricultural products. Valencia is one of the largest intensive cultivation areas of the Iberian Peninsula, so it was used as an indicator to identify some of the most employed pesticides in the country. A study that analyzed 345 samples detected around 40 pesticides in the atmosphere, of which about 37% were insecticides and 33% were fungicides. Regarding insecticides, omethoate displayed a higher incidence, with a detection frequency of 56% and concentrations that reached up to 4 µg/m^3^. Within 18 detected insecticides, bifenthrin, buprofezine, chlorpyrifos, methyl chlorpyrifos, diazinon, dimethoate, imidacloprid and pyriproxyfen were the most prevalent ones. In the case of fungicides, the most common was carbendazim (47% detection frequency), with an average concentration of 140 pg/m^3^. Although the use of this fungicide is not currently allowed, it was not legally prohibited at the time of this study. After carbendazim, the fungicide tebuconazole, from the triazoles family, was found in significant amounts and showed maximum quantities of 7 µg/m^3^. In addition, other fungicides detected in the atmosphere were chlorothalonil, diphenylamine, fludioxonil, folpet, imazalil, iprodione, penconazole, pyrimethanil, thiabendazole and tricyclazole. Regarding herbicides, chlorpropham and carbofuran (among nematicides), whose utilization is currently prohibited in Europe, were detected in the same study [87]. In the following subsections, the agricultural products mainly cultured in Spain will be analyzed in terms of the most detected pesticides (Table 2 and Table 3) which can be classified into five chemical groups (Figure 3).

### 3.2. Potatoes

Potato cultivation requires the application of different pesticides since they can be easily affected by insects, fungus, viruses and weeds. The most used ones in the control of potato cultures in the EU are the insecticide deltamethrin, the herbicide rimsulfuron and the fungicide metalaxyl [88]. Furthermore, chlorpropham is frequently used as a herbicide to inhibit post-harvest outbreaks in stored potatoes, which can produce cross-contamination issues in other stored products, as demonstrated in the UK [89]. However, the EFSA concluded that short- and long-term intake of chlorpropham residues due to this cross-contamination in potatoes is very unlikely to represent a risk for consumer health. The EFSA suggested a temporary maximum level for this residue of 0.3 mg/kg [90]. In Galicia, a region of Spain in the northwest of the Peninsula, the risk of potato infection and disease is high due to the high environmental humidity and the abundance of precipitations, especially in spring. Therefore, in this region, the use of carbofuran and fenamiphos as a treatment for insecticidal and nematocidal pests, as well as metribuzin (herbicide) and deltamethrin, has been very frequently reported. On the other hand, metalaxyl and folpet are usually utilized for their fungicidal properties [91]. Metalaxyl was found in tubers with a concentration of 0.022 mg/kg (Table 2) [88]. Likewise, potatoes collected from different areas of Valencia also showed high amounts of chlorpropham (3.60 mg/kg) compared to other pesticides such as carbendazim (0.01 mg/kg), chlorpyrifos (0.17 mg/kg) or fenoxycarb (0.05 mg/kg) (Table 2) [92]. In EFSA reports carried out on an annual basis, potatoes were targeted in 2014, 2017 and 2020. By 2020, thiram (dithiocarbamate), cypermethrin and dimethoate were detected in potatoes. Regarding the determination of cypermethrin, the highest detected concentration did not exceed the current MRL (Table 3). In the case of dimethoate, despite being detected in just one sample, it contributed to the total chronic exposure with 11.5% of the Acceptable Daily Intake (ADI). The presence of dimetoate should not be detected in potatoes in further reports [74]. In fact, its presence was not described in the 2021 EFSA report [93]. Some other unapproved pesticides were fipronil and chlorpyrifos. In 2020, the exceedance rate of the acute reference dose (ARfD) for potatoes (0.8%) was lower than in 2014 and 2017 (~1.2%) [74]. Future determinations are desirable to continue this trend.

**Table 2 foods-12-03054-t002:** Some detected pesticides in Spanish agricultural crops that have exceeded maximum residue levels (MRL).

Crops	Pesticide Class	Pesticide Detected	MRLs (mg/kg)	Highest Detected Concentration (mg/kg)	Ref.
Potato	Insecticide	Deltamethrin	0.3	-	[88]
		Chlorpyrifos	0.01	0.170	[92]
		Fenoxycarb	0.01	0.050	[92]
	Herbicide	Rimsulfuron	0.01	-	[89]
		Chlorpropham	10	3.600	[92]
	Fungicide	Metalaxyl	0.02	0.022	[88]
		Carbendazim *	0.1	0.010	[92]
Tomato	Insecticide	Fenitrothion	0.01 **	-	
		Chlorpyrifos	0.1	0.730	[94]
		Metidathion *	0.02 **	-	[94]
		Diazinon	0.01 **	-	[94]
		Dimethoate	0.01 **	0.130	[94]
	Fungicide	Carbendazim *	0.3	0.400	[92]
Grapes	Herbicide	Fluometuron	0.01 **	0.174	[95]
		Terbutylazine	0.1	0.403	[95]
	Fungicide	Metalaxyl	2	0.011	[95]
		Triadimenol	0.3	0.026	[95]
		Carbendazim *	0.3	0.290	[92]
	Insecticide	Bifenthrin	0.3	0.080	[92]
		λ-cyhalothrin	0.2	0.07	[92]
		Chlorpyrifos	0.01	0.300	[92]
Oranges	Fungicide	Carbendazim *	0.2	-	
		Thiabendazole	7.0	14.1	[73]
		Imazalil	4.0	12.8	[73]
	Insecticide	λ-cyhalothrin	0.2	-	
		Carbofuran *	0.01 **	-	
		Chlorpyrifos	1.5	-	
Strawberries	Fungicide	Carbendazim *	0.1 **	0.100	[92]
		Tiabendazol	0.05 **	-	
		Imazalil	0.05 **	-	
		Thiophanate-methyl	0.1 **	0.100	[92]
	Insecticide	λ-cyhalotrin	0.01 **	-	
		Carbofuran	0.05 **	-	
		Formethanate	0.05 **	0.470	[92]
		Fenoxicarbp	0.05 **	0.150	[92]
Peppers	Insecticide	Bifentrin	0.5	0.190	[92]
		λ-cyhalothrin	0.1	0.080	[96]
		Cypermethrin	0.5	0.400	[96]
		Acrinathrin	0.02	0.600	[96]
	Fungicide	Thiophanate-methyl	0.1	0.360	[92]
Olive	Fungicide	Chlorpyrifos	0.01 **	-	[73]
Oil		Iprodione	0.01 **	-	
		Chlorothalonil	0.01 **	-	
	Insecticide	Cypermethrin	0.05 **	-	
Cereals	Insecticide	Deltametrin	1.00	2.000	[73]
(rice)	Fungicide	Isoprothiolane	6.00	-	
		Carbendazim *	0.01 **	-	

* prohibited substances, ** lowest detection limit.

### 3.3. Tomatoes

Tomatoes are a basic product in Spanish horticulture and present an enormous demand for exportation to other European countries. Fungal diseases limit their commercialization due to the malformations they produce, which ultimately generate devastating economic losses [97,98]. In addition, fungal infections can occur along all the stages of development of the tomato, even during post-harvest. Mildew caused by the fungus *Phytophthora infestans*, present in many tomato zones, is one of the most catastrophic diseases affecting this cultivation [97]. Tomato cultivation is also affected by viruses transmitted by vector insects [97]. The organophosphate insecticides fenitrothion, chlorpyrifos, methidathion, diazinon and dimethoate are widely used in tomato cultures in Spain. Their concentrations have been described to oscillate between 130 µg/kg (dimethoate) and 730 µg/kg (chlorpyrifos) [94], values that are significantly lower than the established MRL of 0.01 mg/kg (Table 2) [99,100]. From the above-listed insecticides, the levels found for methidathion were the lowest, while the concentration of dimethoate was the highest. In addition, other pesticides frequently present in tomatoes are carbendazim, dietofencarb, fenoxycarb and methyl thiophanate. Among them, carbendazim was the most detected insecticide, in concentrations that may reach values up to 0.40 mg/kg (Table 2) [92]. In this sense, a study concluded that boiling tomatoes may minimize the presence of several pesticides found in this matrix such as fludioxonil (reduced up to 69%), while the application of a bleaching step may significantly contribute to reducing pyridaben residues [101]. Tomatoes have been targeted to be included as a food commodity for the EU-coordinated control program in 2016 and 2019. The non-approved pesticides included chlorfenapyr, triadimefon, acephate, fipronil and permethrin. EU-approved pesticides found in tomatoes included dimethoate, dithiocarbamates, chlorpyrifos and acetamiprid (Table 3). In general terms, outcomes from 2019 regarding MRL exceedance rates for tomatoes displayed a descending trend from 2.6% to 1.7% [102].

Although tomatoes were not selected as food commodities in the annual EFSA reports of 2020 and 2021, different outcomes can be extracted. Chlorates exceedance was observed in these reports; nevertheless, it seems to follow a decreasing tendency. Chlorates are not approved in the EU as pesticide; however, their presence is associated with the use of chlorine-based sanitizing and disinfection solutions required to ensure hygienic conditions in the food industry. Another non-approved pesticide detected, chlorfenapyr, has no import tolerance [74,93]. Among the pesticides with a higher frequency of detection, spinosad and bromide ions were found. Tomatoes were the major food products that contributed to the total chronic exposure of bromide ions. However, it was undetermined if its presence was due to its natural occurrence or to the use of pesticides [74]. In the EFSA report from 2021, tomatoes were mentioned several times, since a few pesticides were identified in this matrix (Table 3) [93].

### 3.4. Grapes

The impact of diseases in vineyards, especially those mediated by fungi, has been recognized as one of the main causes of economic losses in the wine sector. Nevertheless, the affection of vineyards implies losses for other industrial food sectors, since grapes can be consumed both fresh and processed as jam, juice, jelly, oil of grape seeds, raisins and vinegar. The most habitual fungal diseases in vineyards all over the world are gray molds (*Botrytis cinerea*) and powdery (*Uncicula necator*) and downy mildews (*Plasmopara viticola*) [103]. A study on pesticides in the soil of seven vineyards in the Spanish region of La Rioja detected the presence of the highest concentrations of the herbicides fluometuron and terbuthylazine, followed by the fungicides metalaxyl and triadimenol and the insecticide methoxyphenozide (Table 2) [95]. Another study that analyzed fruits and vegetables from Valencia demonstrated that the highest concentration of pesticides in grapes corresponded to chlorpyrifos and carbendazim, followed by bifenthrin and λ-cyhalothrin (Table 2) [92]. However, grapes from Murcia displayed much higher levels, such as in the case of flufenoxuron, the concentration of which was the highest, with a value of 0.57 mg/kg, followed fenoxycarb (0.27 mg/kg), pyriproxyfen (0.18 mg/kg) and lufenuron (0.11 mg/kg) [104]. In a work published in 2022, notifications on pesticide residues in food submitted to the Rapid Alert System for Food and Feed (RASFF) were analyzed for the period between 2002 and 2020. This study reported grapes as one of the most concerning products, since 17% of the notifications relative to fruits were related to this product. The annual EFSA report of 2021 included as food commodity grapes that contain a rate of multiple residues of 22%. Indeed, in one sample of table grapes from Turkey, up to 19 different pesticides were identified. Regarding ARfD exceedances, table grapes contributed 91 samples that contain dithiocarbamates (ziriam, maneb, mancozeb, propineb and thiram), cyhalothrin (gamma-cyhalothrin was detected in 12 samples), cypermethrin (11 samples), acetamiprid (23 samples), indoxacarb (19 samples) and omethoate (1 sample) (Table 3) [93]. Regarding cyhalothrin, the quantification of isomers was included, although their differentiation is not performed routinely. To better protect consumers, the EFSA conducts risk assessments considering the most toxic isomer, gamma-cyhalothrin. An adequate candidate substitute for cypermethrin is being sought. In addition, lowering the MRL is another measure under consideration to minimize consumer exposure. In general terms, the comparison of exceedance outcomes from 2021 (2.1%) against those from 2018 (2.6%) displayed a decreasing tendency for table grapes. However, exceedance rates from 2018 and 2021 are still higher than those from 2015 [93].

### 3.5. Oranges

The most frequently found pesticides in different citrus (oranges, mandarin or lemons), assessed in several Valencian products, were chlorpyrifos, methyl chlorpyrifos and carbendazim [92]. Indeed, the annual report of the EFSA from 2021 showed the presence of chlorpyrifos, chlorpyrifos-methyl, buprofezin and prochloraz in 27, 44, 16 and 7 grapefruit samples from Turkey, respectively [93]. Another work compared pesticides present in essential oils obtained from oranges produced under ecological and conventional models. Essential oils from conventionally cultured oranges showed the presence of 56 different pesticides, 28 of them were frequently detected (≥10 times) and the most relevant ones were chlorpyrifos, diphenylamine and biphenyl. Pesticide concentrations (>10 μg/L) surpassed MRL levels frequently (72%). For example, the quantification of imazalil and propargite was 15 mg/L, for prochloraz it was 16 mg/L, and for pyrimethanil it was 19 mg/L. Since these pesticides were mainly detected in orange skins, different strategies were assessed to minimize their concentrations. Washing, squeezing or pasteurization were suggested to be the most efficient approaches to reduce their presence in juice. In the case of essential oils obtained from organic orange samples, 18 pesticides were detected. Among the pesticides found, diphenylamine, biphenyl, chlorpyrifos and atrazine were identified even though their concentrations were much lower than in conventionally produced oranges. In fact, detected concentrations ranged from 100 to 278 μg/L for piperonyl butoxide [101]. Regarding EFSA reports, oranges were selected as a reference fruit for the development of the assessment of citrus (oranges, lemons, limes, mandarins, etc.). Oranges were targeted as food commodities in the consecutive reports performed in 2014, 2017 and 2020. In 2017, λ-cyhalothrin (one sample) and dimethoate (seven samples: three European (Italy, Malta, and Spain), two from Egypt, and two from Lebanon) were detected. Imazalil was identified in 1150 samples and, although only one exceeded the MRL (12.8 mg/kg, Table 2) it represented 238% of the ARfD. The highest concentrations of imazalil were quantified in the skin, which represents a key point when analyzing this information in the context of the reutilization of orange residues. Thiabendazole exceeded the MRL in six samples with an ARfD of 318%. Other disallowed pesticides identified in oranges above the MRL and ARfD values included carbendazim (two samples from Argentina); carbofuran (one sample, from Spain); and fenthion (two samples from Malta and Spain). In addition, unapproved pesticides in Europe were identified, such as methidathion, chlorfenapyr and profhenophos [73]. The EFSA report from 2020 indicated that 762 samples of oranges represented the highest number of samples with multiple residues [74]. In this sense, the work that analyzed notifications on pesticides submitted to RASFF (2002–2020) also indicated that citrus was one of the fruit categories that was most often highlighted (306 notifications), with oranges as the most notified product (117 notifications) [69]. Indeed, 28 samples grown in the EU and 27 grown in non-EU countries contributed, with a total of 16 non-compliant samples. Other approved pesticides found in oranges in 2020 included cypermethrin, dimethoate, dithiocarbamates (maneb, mancozeb, propineb, thiram and ziram), thiabendazole and omethoate. In summary, the trend in ARfD for oranges (2.9%) shows an increase in 2014 (1.5%) and in 2017 (1.1%).

### 3.6. Strawberry and Other Berries

Strawberries are cultivated in the same soil every year without rotation with other crops [105]. Consequently, there is a constant presence of pathogens and nematodes that can attack their production. Indeed, the presence of nematodes in non-fumigated fields can be up to 10 times higher than in fumigated ones [106]. Spain is the most important producer of strawberries in Europe and the sixth in the world. Specifically, Andalusia is the region with the highest area of strawberry cultivation which mainly uses protective structures such as greenhouses and plastic tunnels [34]. These protective structures have the main objective of providing year-round fruit availability. To this aim, strawberry cultivation is required to be resilient against adverse weather conditions and pathogens. In this sense, the control of fungus and weeds in protective structures is easier but still conducted with insecticides, fungicides, acaricides and herbicides such as cyprodinil, fosetyl aluminum, penconazole, abamectin, glufosinate-ammonium and bupirimate [34]. The work analyzing the notifications of pesticide residues in food submitted to the RASFF (2002–2020) showed that the most concerning pesticide residue notifications within the category of fruits were relative to berries and small fruits (345 notifications), of which strawberries accounted for 82 notifications. More specifically, the previously mentioned study of Valencian agricultural products highlighted the presence of carbamates in strawberries. Maximal detected levels were observed for formethanate, phenoxycarb, carbendazim and methyl thiophanate (Table 2) [92]. A contemporaneous study assessed the efficiency of several approaches to prevent the intake of these residues. The procedures included washing using tap or ozonated water, boiling or ultrasound-based cleaning. The latter two strategies significantly reduced the content of pesticides in strawberries up to 91–92%. Regarding the efficiency of the approaches, ozonated water washing reduced the pesticide presence by 36–75%, whereas tap water washing reduced the presence of bupirimate, λ-cyhalothrin, fludioxonil, cyprodinil and methyl chlorpyrifos to levels by between 20 and 68%. Therefore, tap water washing seems a much more scalable and affordable option since it does not require the use of instruments to ozonate the water [107]. 

Regarding other berries, significant differences were observed among the national legal frameworks. For instance, pesticides such as bifenthrin, clothianidin, imidacloprid, fenpropathrin, methomyl, thiamethoxam and zeta-cypermethrin are commonly used in the USA for culturing raspberries, blackberries and blueberries. On the contrary, in 2013, the EU applied several restrictive measures for the use of clothianidin, imidacloprid and thiamethoxam, since they were described as harmful pesticides for bees and thus honey production may be badly affected [108]. In 2018, in the annual EFSA report, Goji berries from China were found to contain several pesticides including anthraquinone, carbofuran and nicotine. Indeed, an individual goji berry sample contained up to 29 different pesticides. Additionally, black, red and white currants presented the highest frequency (76%) of multiple residues of the total unprocessed samples [109]. In the same line, in the EFSA report from 2019, several berries were underlined to have the highest frequency of multiple residues; black, red and white currants contained 72.6%, sweet cherries contained 69.2% and strawberries contained 63.6%. In fact, six samples of strawberries were highlighted for exceeding some pesticide levels. Among the approved pesticides found in strawberries that surpassed both the ARfD and the MRL was abamectin. Other identified pesticides included acrinathrin, captan, chlorpyrifos, fenamiphos, formetanate, oxamyl and tebuconazole, among others. Regarding non-approved substances below the MRL, acephate, antraquinone, carbendazim, chlorates, chlorpropham, dimethoate and omethoate, iprodione, methomyl, triadimenol and pymetrozine were found, among others (Table 3). In general terms, when comparing results from the 2016 report against 2019, the individual MRL exceedance rate increased for strawberries from 1.8% to 3.3% [102].

### 3.7. Peppers

Sweet peppers, together with tomato, are the most important greenhouse horticultural cultivation in the southeast of Spain, specifically in Almeria. In fact, sweet pepper production has reinforced the economic engine of this province in recent decades [110,111]. Among the prevalent diseases that may affect the production of peppers in greenhouses, the viral infection transmitted by the insect *Frankliniella occidentalis* is the most problematic one. Tomato Spotted Wilt Virus has become one of the most harmful plant viral pathogens due to its resistance to several active compounds [111]. This resistance has started endless feedback that has led to an increase in the use of pesticides to combat it. Since the virus is transmitted by trips, some of the most used pesticides are insecticides such as emamectin benzoate, sulfoxaflor, spirotetramat, pymetrozine, chlorantraniliprole, bifenthrin and cyfluthrin, but the fungicide carbendazim has also been found in peppers [92,110]. The insecticide bifenthrin was distantly followed in concentration by cyfluthrin and carbendazim (Table 2) [92]. In addition, a recent study analyzed the levels of the pyrethroid insecticides in peppers acquired from Almeria markets. The outcomes demonstrated the presence of acrinathrin, cypermethrin and λ-cyhalothrin (Table 2) [96]. Regarding the annual reports of the EFSA, in 2018, the presence of fipronil in spicy peppers from the Dominican Republic was revealed. Following this identification, the EFSA suggested the inclusion of this pesticide in future analyses of fruits and vegetables [109]. The random analysis of peppers in the EFSA report of 2020 revealed the presence of chlorpyrifos and chlorpyrifos-methyl, two non-approved pesticides with no import tolerance. The last EFSA report where peppers were selected as individual food commodities was in 2021 (the previous ones were in 2015 and 2018). Peppers were highlighted as one of the food products with a high rate of multiple residues with a frequency of 12.8%. Indeed, just one sample from Cambodia contained up to 28 different pesticides. In this line, 34 samples of sweet/bell peppers showed ARfD exceedances. These exceedances were observed for dithiocarbamates, cyhalothrin, cypermethrin, acetamiprid, indoxacarb, bromide ion, omethoate and oxamyl. In summary, MRL exceedance rates for peppers increased in the three reports from 2015 (0.8%), 2018 (2.4%) and 2021 (3.4%). Indeed, the potential presence of pesticides in peppers was notified 876 times to the RASFF between 2002 and 2020, which indicates that peppers were the most often concerned vegetable in this regard [69].

### 3.8. Olives

The extensive use of pesticides in olive groves has led to the development of validated and specific analytical methods to detect them both in olives and olive oil. Indeed, as olive oil is a product obtained after olive processing, it represents a good model to identify the potential presence of pesticides in the residues generated by this industry. The approved analytical methods to detect pesticides in vegetal oils and other by-products have been published in Regulation (CE) no. 1107/2009, related to the commercialization of phytosanitary products [112]. The presence of pesticides in olives aimed to produce olive oil is regulated under MRLs specified in Regulation (CE) 396/2005, whose application allows the determination of 472 pesticides [80]. In a previously mentioned work that analyzed pesticide composition in several products and the surrounding aerial environment, the pesticides present in the atmosphere associated with the culture of olives were omethoate and dimethoate. In fact, omethoate was frequently found at levels of 30 µg/m^3^ in rural areas, while dimethoate was detected close to olive crops at concentrations between 0.3 and 17 µg/m^3^, depending on the analyzed area [87]. These pesticides were not detected in the EFSA reports of 2015, 2018 or 2021 for olive oil but they were present in other food commodities, which may indicate the presence of cross-contamination due to environmental pesticide spread. The EFSA report of 2018 highlighted four olive oil samples from the main producer countries (two from Cyprus, one from Italy and one from Spain) that contained pesticides above the MRLs. These pesticides were cypermethrin, iprodione, chlorpyrifos and chlorothalonil. [109]. Iprodione and chlorpyrifos were commonly found in 2015 and 2018. The presence of chlorpyrifos had already been confirmed in the report in 2012, together with terbuthylazine and other pesticides that exceeded the MRL (Table 3). However, since a correction factor of 5 was required for the final calculation of the MRL of these samples, these mentioned pesticides were not considered to be out of the legal limits [113]. The last EFSA report in which olive oil was included as a food commodity was in 2021 and no references to pesticide presence was documented. Indeed, a decreasing tendency was noticed for virgin olive oil regarding the exceedance rates from 2015 (0.9%) to 2018 (0.6%) and to 2021 (0.3%).

**Table 3 foods-12-03054-t003:** Summary of EFSA reports of pesticides performed on an annual basis regarding major Spanish agricultural crops.

Crops	EFSA Report	Pesticide	Concentration/Status	Notes	Refs.
Potato	2020	Dithiocarbamate (thiram)		Approved isomers: alpha and zeta	[74]
		Dimethoate	Not approved	Grace period: June 2020	
		Cypermethrin *	0.039 mg/kg		
		Fipronil, chlorpyrifos	Not approved,	Non-compliant sample	
Tomato	2019	Chlorfenapy	>MRL. Not approved	Origin: EU and non-EU counties	[102]
		Triadimefon	>MRL. Not approved	Origin: EU countries	
		Acephate, fipronil, permethrin	Not approved	Origin: non-EU counties	
		Dimethoate		Still approved (8 samples)	
		Dithiocarbamates (ziram, maneb, propineb and thiram)	<MRL		
		Chlorpyrifos	≤MRL	ARfD exceedance 115%	
		Acetamiprid		ARfD exceedance	
	2020	Chlorates	>MRL (10 samples) Not approved as pesticide	Decreasing tendency	
	2020/2021	Chlorfenapyr	Not approved	No import tolerance	[74,93]
		Bromide ion		Total chronic exposure: 8.1% ADI	
		Spinosad		High frequency of detection (5.6%)	
	2021	Abamectin, oxamyl, phosmet and dithiocarbamates (thiram)		Not targeted as food commodity	[93]
Grapes	2021	Cyhalothrin **		Grace period: October 2022 Non-compliant: 2 samples (Cyprus)	[93]
		Acetamiprid	>MRL: 0.34–0.81 mg/kg	19 samples	
		Indoxacarb	<MRL Approval not renewed	Grace period September 2022	
		Omethoate	Never approved	Non-compliant sample (Cyprus) Mutagenic	
Oranges	2020	Dimethoate, chlorpropham and linuron *	Not approved.	Non-compliant samples. Origin: EU countries	[74]
		Bromopropylate, fenbutatin oxide, carbendazim, profenofos	Not approved.	Non-compliant samples. Origin: non-EU countries	
		Cypermethrin *	0.12 mg/kg Not approved	ARfD and low toxicology: consideration for processing factor application	
		Dimethoate	>MRL (13 samples)	Grace period: 30 June 2020. Total chronic exposure: 19% ADI EFSA’s suggestion: keep monitoring	
		Dithiocarbamates		Exceedances rates: when illegal use	
		Omeoathe	Never approved (3 samples)	Mutagenic	
		Thiabendazole	>MRL (3 samples) >ARfD (9 samples)	Applied a peeling factor of 0.17	
Berries	2018	29 pesticides (1 Goji sample)		Highest frequency of multiple residues	[109]
	2019	Carbofuran	>MRL Non approved.	Origin: EU countries	[102]
		Dichlorvos	>MRL Non approved	Origin: other counties	
Peppers	2021	Dithiocarbamates	>ARfD	Presence of precursors (ziram, propineb or thiram)	[93]
		Cyhalothrin **	>ARfD (8 samples)	Gamma isomer not authorized Approval expiration October 2022	
		Acetamiprid	>ARfD 0.56 mg/kg 0.61 mg/kg	>ARfD (3 samples) >MRL (2 samples)	
		Indoxacarb	>ARfD/<MRL	Grace period: September 2022	
		Bromide ion		119 positive samples	
		Omethoate	Never approved	Non-compliant samples (Uganda, Morocco) Mutagenic	
		Chlorfenapyr	Not approved	Origin: non-EU countries (Cambodia, Albania)	
		Ethephon		5 positive samples (3 Poland, 2 Spain, 1 The Netherlands)	
Olives	2012	Chlorpyrifos and terbuthylazine		Chlorpyrifos in 14% of samples, terbuthylazine 12%	[113]
		Endosulfan, famoxadone, pendimethalin, fenthion, and terbuthylazine	>MRL	Highly frequent: fenthion and terbuthylazine	
	2015	Bromopropylate, chlorpyrifos, methyl-chlorpyrifos, iprodione, and fenthion	-		[109]
	2018	Cypermethrin *, iprodione, chlorpyrifos, and chlorothalonil	-		[109]
		Chlorothalonil	-		
		Cypermethrin *	-		
Cereals	2017	Deltametrin	>ARfD (1.7 mg/kg)	2017 MRL 2 mg/kg Current MRL 1 mg/kg	[114]
(rice)		Isoprothiolane, carbendazim	>MRL	Origin: EU countries Carbendazim: not approved	
		Acephate, hexaconazole, methamidophos, triazophos	>MRL	Origin: non-EU countries	
	2020	One sample: 15 pesticides		134 multiresidue positive simples	[74]
		Thiamethoxam	Not approved	Origin: EU countries	
		Tricyclazole, hexaconazole, thiamethoxam and chlorpyrifos	Not approved	Origin: non-EU countries	
		Bromide ion	>ARfD	Total chronic exposure: 5.8% ADI	

* Sum of four enantiomers: alpha, beta, theta and zeta substances, ** Sum of two isomers: lambda and gamma.

### 3.9. Cereals

In the EFSA report of 2017, rice received special attention, since multiple pesticides were identified during that same year. Deltamethrin exceeded the ARfD limits in three samples. Between 2014 and 2017, rice of European origin showed a marked trend for certain pesticides that exceeded MRLs, such as in the case of isoprothiolane or carbendazim. For this reason, in light of the results of this period, the EFSA recommended verifying the content of isoprothiolane, bromide ions, propiconazole, deltamethrin, tebuconazole, buprofezine, imidacloprid, carbendazim and thiamethoxam [73]. In Spain, the national report relative to 2018 and provided to the EFSA indicated that only one rice sample contained just one type of pesticide, tricyclazole [114]. The last EFSA report where rice was included as a food commodity was in 2020, where 134 samples displayed the presence of multiple residues. In fact, a sample of rice that contained up to 15 different pesticides lead to two non-compliant samples. In addition, several positive results were detected for samples from Pakistan such as three for carbendazim, one for profenofos and one for triazophos. In 2020, rice presented two samples with ARfD exceedances for bromide ions, despite having an origin from natural sources [74]. The presence of bromide ions, carbendazim, hexaconazole and triazophos was commonly detected in rice in 2017 and 2020. In addition, the detection of seven non-approved pesticides has led to the recommendation of maintaining a monitoring program for rice. Indeed, rice has presented a concerning increasing MRL exceedance rate over the years, with values of 5.1% in 2017 and 6.7% in 2020 [74].

On the other hand, regarding other cereals, permethrin was detected in rye of European origin in the EFSA report from 2017, while in 2018, carbendazim was frequently found in wheat. In addition, fenitrothion appeared in two samples, both of European origin [73,109]. Finally, in the EFSA report from 2020, wheat was estimated to contribute to the total chronic exposure of bromide ions with 38.8% ADI [74].

## 4. Adverse Effects of Pesticides on Human and Animal Health

In recent decades, the use of substances that prevent the insect attacks, microorganisms and weed plagues has become a necessity to achieve the quality required by markets and/or the economic efficiency of crops. However, many compounds used for this purpose may become a risk factor for human health if they are not properly managed. The consumption of pesticides through vegetal matrixes treated with the above-named compounds may generate toxicity both in humans and animals. They usually accumulate in fatty tissues, and when high amounts bioaccumulate they may decrease life quality and be responsible for the development of fatal affections [115]. For this reason, the EFSA has suggested the development of models to evaluate risk. The “pre-marketing” model aims to evaluate the risk of a new pesticide or to add a new use to an already used substance, whereas the purpose of the “post-marketing” model is related to the evaluation of real exposition of consumers to pesticides in foods. Both methodologies have been developed to provide two temporary frame evaluations: a short-term exposure (acute), where ARfD data are used, and a long-term exposure (chronic), based on ADI data [61].

As demonstrated in different kinds of works, generally, pesticides are easily distributed in the environment [60,62,87]. Therefore, humans may be exposed to them through food and/or contaminated water by different exposition ways such as respiration and inhalation, skin and mucosal contact or ingestion [115]. Within an organism, these compounds can accumulate in fatty tissues or in corporal fluids with a marked lipidic nature; indeed, the presence of organic pollutants has already been detected in maternal milk, leading to postnatal exposure during breastfeeding [60]. Cases of acute intoxication have been described to produce symptoms such as headaches, dizziness, nausea, vomiting, muscle spasms, seizures, anxiety and confusion, while chronic exposure is related to severe effects such as cerebrovascular and liver diseases, reproductive and/or nervous affections and even cancer [62,115]. For all these reasons, it is crucial to understand and identify health risks that pesticides may induce in humans, animals and in different kinds of environments.

The most widely used pesticides in Spain are classified into five chemical groups (Figure 3), each of them related to specific adverse effects.

Organochlorines are considered the most toxic group, which in addition present wide environmental dispersion and long-term persistence. Short-term adverse effects induced by organochlorines in human health include convulsions, headaches, nausea, trembling, spasms, muscular weakness and speech difficulties. The long-term effects affect the correct function of the liver, kidneys, bladder, thyroid and central nervous system [116]. Imidacloprid, chlorothalonil, folpet, iprodione and penconazole are some representative examples of the group of organochlorine pesticides.

The group of organophosphates is extensively used to control crop infections mediated by vectors. Acute exposure to organophosphates mainly involves the respiratory and digestive tract, causing bronchospasms, pulmonary edema, rhinorrhea, nausea, headaches, dizziness, diarrhea and vomiting. Long-term exposure to organophosphates may cause several systemic disorders such as muscle paralysis or systemic failures such as respiratory arrest or neurotoxic effects [117]. Among the most used organophosphates in agricultural products in Spain are diazinon, chlorpyrifos and dimethoate. The insecticide chlorpyrifos and its methylated derivative have been classified as harmful agents for human health for being toxic to reproduction, category 1B [63,64].

Regarding carbamates, another category of pesticides, those mainly utilized in Spain include chlorpropham, carbendazim and carbofuran. Acute exposure to carbamates produces muscular weakness, dizziness, salivation, headaches, nausea, diarrhea and vomiting. Long-term contact with carbamates was described to induce neuro-psychologic sequelae and to induce carcinogenic and mutagenic effects [118]. In March 2015, carbendazim was suggested to be substituted, since it was classified as category 1B toxic to reproduction and as a category 1B mutagenic [65]. Nowadays, the application of carbendazim is not allowed in Europe [65].

The chemical group of synthetic pyrethroids has been reported to cause muscle fasciculations like facial paraesthesia, skin itching or burning, dizziness, nausea or vomiting [119] after short-term exposure, and they may have toxic effects related to reproductive toxicity, immunotoxicity, endocrine disruption, hepatotoxicity, cardiotoxicity and neurotoxicity when exposure is prolonged [66]. Among the variety of chemical structures of this pesticide family, bifenthrin is one well-known representative.

Finally, the group of triazines, such as buprofezine, are the most used herbicides in agricultural cultivation. Despite the fact that they have been utilized to substitute the application of some organochlorines, triazines have also been associated with several acute adverse effects. Mild symptoms include abdominal pain, dermatitis, diarrhea, nausea and eye irritation; however, it has been pointed out that long-term exposure to triazines may be involved in cancer development, teratogenesis and hormonal disorders [115]. However, the EFSA indicated that triazine amine, a common metabolite formed during the metabolism and breakdown of triazinylsulfonylurea compounds, has no potential to induce gene mutations and clastogenicity [67].

Even though pesticides are used to eliminate, prevent or control the negative impact of undesirable species in crops, their lack of specificity represents a threat to other animal groups [120]. Animal exposure to pesticides can generate long-term adverse effects similar to those induced in humans (carcinogenesis, immunotoxicity, endocrine disruption, obesity, reproductive failure, brain development disorder and behavior alterations) in addition to a range of acute effects [59,121]. As in humans, the most common sources of pesticide exposure include inhalation of contaminated air or the ingestion of contaminated feed products [59,121]. Continuous exposure to pesticides of lipophilic nature such as organophosphates, organochlorines, carbamates and pyrethroids can lead to potential bioaccumulation in fatty tissues [121]. Among them, carbamates and organophosphates seem to be the main pesticides involved in common intoxication cases of both domestic and wild animals [59,122].

Accidental intoxications of animals, apart from affecting animal wellbeing, can create other public health issues derived from human intoxication after consuming affected animals. On the other hand, the environmental persistence of pesticides promotes their geographical dispersion towards distant ecosystems where they can still affect the fauna [123]. A few persistent pesticides have been described to remain for long periods of time. For instance, dichloro-diphenyl trichloroethane is still detected in the environment, although its use has been prohibited in the USA since 1970 [124]. A closer example is carbofuran, a pesticide of the carbamate family whose application was banned in 2007, which is still detected in animals in Italy [68,122].

Currently, the use of pesticides is tightly controlled at an international level. The establishment of MRLs minimizes their contact with humans. In fact, these applied levels do not exceed the ARfD for consumers. Controlled application and adequate analysis of products before they reach markets are two key tools to prevent the exposure to high concentrations associated with health risks [125]. However, more studies are needed to deeply analyze the process of bioaccumulation, the potential synergies and antagonisms of mixtures of pesticides and how they affect human and animal health.

## 5. Safety Assessment of Citrus Peels as a Matrix to Recover Ascorbic Acid: Case Study

A safety assessment of the revalorization of citrus fruits peels to recover vitamin C is presented in this section. The research stages to perform this safety assessment are presented in Figure 2, whereas Table 4 summarizes the values obtained after the application of each research step.

The first step was a search for the content of ascorbic acid (AA) in peels of several citrus fruit species. The outcomes are presented in the ‘AA’ column of Table 4. A wide variability of AA concentrations was observed depending on the citrus species used but also on the extraction technique or solvent used. The content of AA in peels from oranges, *Citrus sinensis*, ranged from 1.355 to 136 mg/100 g of dw and 8.89 to 93.33 mg/100 g of fw (second column of Table 4). As observed, the AA concentration was reported on a fresh weight (fw) or dry weight (dw) basis, depending on the selected work. The results of AA concentration were homogenized to fresh weight using a ‘water content’ factor of 6.6X (third column of Table 4). This conversion was necessary since residue determination is quantified using fresh weight products. Then, two values of peel biomass weight were obtained. These weights represent the amount of peel needed to recover enough AA to fulfill the lowest RDA (45 mg, fourth column of Table 4) or the highest RDA (200 mg, fifth column of Table 4). The highest weight of peel would be required for two samples of *C. sinensis* and one sample of *C. latifolia*, since their AA content is lower than 0.09 mg/g. Then, the hypothetical presence of imazalil and thiabendazole at concentrations equal to their respective MRLs was considered (4 or 5 mg/kg for imazalil and 7 mg/kg for thiabendazole). In this way, the hypothetical amount of these two pesticides was calculated for all the citrus peels. The amount of thiabendazole in the biomass required to fulfill 45 mg of AA (sixth column in Table 4) or 200 mg of AA (eight column in Table 4) was considered. Identically, the amount of imazalil in the citrus biomass required to fulfill 45 mg of AA (tenth column in Table 4) or 200 mg of AA (twelfth column in Table 4) was calculated. Then, a processing factor of 45% for thiabendazole and 54% for imazalil was applied (7th, 9th, 11th, 13th columns in Table 4). The results are evaluated considering the acceptable daily intake (ADI), the ARfD of each pesticide [75,76] and a body weight of 70 kg [126]. Hence, the ADI for imazalil is 1.75 mg/kg and the ARfD is 3.5 mg/kg, while for thianbendazole the ADI and ARfD are 7 mg/kg.

As Table 4 shows, the results among pesticides are quite similar even when their respective MRLs are slightly different. The reutilization of citrus peels previously washed with tap water to recover AA for a daily intake of 45 mg is safe in most cases. For thiabendazole, none of the samples would surpass either the ADI or ARfD values. For imazalil, although the MRLs are lower, one sample would surpass the ADI value, although it would be reduced below this limit after the washing step. When looking at results to obtain 200 mg of AA, the data also show nearly total protection of consumers. In terms of exposure, just one sample surpasses the ADI and ARfD values for thiabendazole, even after applying the processing factor. For imazalil, one sample would surpass the ADI limit, but the washing step would satisfactorily reduce the concentration. Another sample exceeds both the ADI and ARfD limits, even after the correction with the washing step (Table 4). If the highest EFSA average requirement for AA is taken into consideration, i.e., 145 mg for women during lactation [127], just one sample would surpass the ADI/ARfD for thiabendazole and imazalil. This sample is the one with the lowest AA concentration. Indeed, samples with AA contents of >10 mg/100 g fw or >1.5 mg/100 g dw would represent safe sources for consumers regarding thiabendazole and imazalil, even when the concentration of pesticides was assumed to be equal to their MRL. Therefore, the reutilization of by-products to recover key nutrients or biomolecules seems to represent a safe option, at least in the scenario of the recovery of a major bio-compound. In this sense, to properly assess the safety of the by-products, it would be necessary to evaluate the relative content of the target biomolecules to establish the biomolecule/by-product ratio. Molecules present at very low levels would require the use of high volumes of by-products, which would imply the accumulation of pesticides. Previously published works have determined the presence of pesticides in by-products derived from processing olives, coffee, oranges, lemons, strawberries, soybeans or cow peas using analytical techniques [128,129,130,131,132,133]. Each study used a different weight of biomass and volume of solvent, in addition to using very variable kinds of solvents. The studies that used lower biomass weights (50 mg or 10 g) did not detect pesticides or they were detected below the MRL [128,130]. The studies that used higher biomass weights (25 g, 500 g or 1 kg) found pesticides above the MRL [129,131,132]. As previously pointed out, the relative biomolecule/by-product ratio is extremely relevant to determine if the revalorization of biomass to recover biomolecules is a safe procedure.

The implementation of pre-treatments to remove the maximal content of pesticides from by-products and the optimization of extraction protocols to recover the highest yield of biomolecules is critical to better protect vulnerable consumers with higher nutritional requirements, such as elderly people, pregnant women or babies during breastfeeding.

Therefore, the reutilization of by-products to recover key nutrients or biomolecules seems to represent a safe option and may be an alternative option to minimize the effect of biomass waste accumulation. However, to truly reduce the environmental impact, the recovery process is required to be performed using efficient extraction techniques that minimize the volume of solvent and maximize the use of biodegradable solvents, such as natural deep eutectic solvents (NADES) [129]. In addition, this waste-to-biomolecules approach cannot represent the only alternative to implement. It must represent one strategy among others such as the production of bioethanol [40] or the exploitation of biomass waste as a matrix to produce single-cell proteins [134] or feeding insects [135]. Both of the latter approaches represent alternative strategies for a proximate future model of food consumption for humans.

**Table 4 foods-12-03054-t004:** Safety assessment of citrus peels as a matrix to recover ascorbic acid (AA). Content of AA found in several species of citrus peels, citrus peels (biomass) weight (g) needed to fulfill daily requirements of 45 or 200 mg AA, associated hypothetical exposure to thiabendazole or imazalil (using MRL) and the application of the processing factor (PF) due to washing procedures (* indicates content of AA per g of fresh weight (fw) biomass otherwise content is based on dry weight. Italics indicate pesticides concentrations above the ADI).

Citrus Peel (Species)	AA (mg/g)	AA (mg/g of fw)	Biomass (g) for 45 mg AA	Biomass (g) for 200 mg AA	Thiabendazole (mg/kg) Exposure				Imazalil (mg/kg) Exposure				Refs.
					45 mg AA/Day	PF	200 mg AA/Day	PF	45 mg AA/Day	PF	200 mg AA/Day	PF	
*C. latifolia*	0.07	0.45	99.68	443.03	0.70	0.31	3.1	1.40	0.50	0.27	*2.22*	1.20	[136]
	0.23	1.49	30.17	134.08	0.21	0.10	0.9	0.42	0.15	0.08	0.67	0.36	[136]
*C. limon*	0.59	3.87	11.64	51.72	0.08	0.04	0.4	0.16	0.06	0.03	0.26	0.14	[49]
	0.26 *	1.75	25.77	114.52	0.18	0.08	0.8	0.36	0.13	0.07	0.57	0.31	[137]
*C. máxima*	0.19 *	1.28	35.25	156.69	0.25	0.11	1.1	0.49	0.14	0.08	0.63	0.34	[138]
*C. paradisi*	1.13	7.48	6.02	26.75	0.04	0.02	0.2	0.08	0.02	0.01	0.11	0.06	[49]
*C. reticulata*	0.48	3.14	14.32	63.66	0.10	0.05	0.4	0.20	0.07	0.04	0.32	0.17	[136]
*C. sinensis*	0.09 *	0.59	76.69	340.87	0.54	0.24	2.4	1.07	0.31	0.17	1.36	0.74	[50]
	0.93 *	6.16	7.31	32.47	0.05	0.02	0.2	0.10	0.03	0.02	0.13	0.07	[50]
	0.43	2.85	15.78	70.15	0.11	0.05	0.5	0.22	0.06	0.03	0.28	0.15	[136]
	0.24	1.60	28.06	124.70	0.20	0.09	0.9	0.39	0.11	0.06	0.50	0.27	[136]
	1.10	7.29	6.18	27.45	0.04	0.02	0.2	0.09	0.02	0.01	0.11	0.06	[49]
	1.36	8.98	5.01	22.28	0.04	0.02	0.2	0.07	0.02	0.01	0.09	0.05	[139]
	0.01	0.09	503.19	2236.39	3.52	1.59	*15.7*	*7.04*	*2.01*	1.09	*8.95*	*4.83*	[48]

## 6. Conclusions

The excessive volume of residues generated by the food industry is directly affected by several factors, among them, trade globalization and the growth in world population are two major actors. Currently, in modern agriculture, tons of residues are burned or accumulate in landfills and thus they threaten environmental ecosystems and public health. The application of a circular economy and a bioeconomy as alternative production models may represent suitable solutions to sustainably manage excess agri-food residues. In Spain, potatoes, olives, tomatoes, grapes and strawberries are the most relevant agricultural products. These residues are rich in bioactive molecules of different nature, such as phenolic compounds, pigments and vitamins. Natural bio-compounds have gained attention as fortification products due to the wide range of functional properties they possess, for example, colorant, antioxidant, antimicrobial and anticancer properties, among others. For this reason, the recovery of these molecules from agricultural residues for further incorporation in food matrixes would lead to a double advantageous strategy. It would reduce waste biomass volume through their revalorization as a source of biomolecules which would represent an economic input for the industrial sector. This work assessed the presence of pesticides in eight major crops in Spain and their adverse effects. The presence of substances not approved for use as pesticides like dimethoate, carbendazim, lambda-cyhalothrin, chlorfenapyr, bromide ions or chlorates generates concern in Europe. Indeed, the EFSA has suggested to keep monitoring them to track their trends. However, other approved pesticides also create concerns when exceeding the MRL or even the ARfD. This is the case for acetamiprid, thiabendazole, imazalil and deltamethrin, among others. Even though their use is approved, excessive exposure to them may represent a risk factor for health. In this sense, the revalorization of biomass to recover biomolecules may lead to an accumulation of pesticides, especially when using peels as a matrix. The safety assessment case study using citrus peels to recover ascorbic acid points to the safety of the revalorization process. However, three key factors need to be considered: the relative content of the target biomolecule, the optimization of the extraction protocols and the application of a washing procedure to decrease the pesticide concentration of the by-products. These factors may represent the key to better protect vulnerable consumers with higher nutritional requirements, such as elderly people, pregnant women or babies during breastfeeding. Nevertheless, to fully minimize the risk of exposure to pesticides, controls conducted by public health organizations where foodstuffs are evaluated for their compliance with MRLs are crucial. These frequent controls ensure that pesticide concentrations are below the harmful limits for health and protect environmental, animal and human welfare.

Therefore, the reutilization of by-products to recover key nutrients or biomolecules seems to represent a safe option and may be an alternative option to minimize the effect of biomass waste accumulation. However, to truly reduce the environmental impact of the recovery process, the use of efficient extraction techniques and biodegradable solvents is required. In addition, the recovery approach cannot represent the only alternative to effectively reduce biomass waste volumes. Other sustainable options include the production of bio-ethanol, single-cell proteins or feed for insect larvae. Therefore, even though the current alternative production models point to many sustainable systems, they still are required to be critically analyzed and applied considering human and animal health in addition to the environmental impact.

## Figures and Tables

**Figure 1 foods-12-03054-f001:**
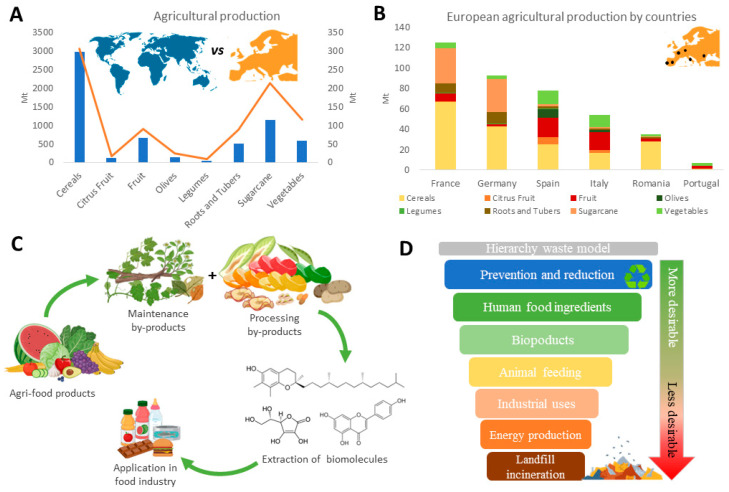
Insights into global and European agricultural production in the context of the circular economy strategy based on the hierarchical waste model. Agricultural production of eight major crops at (**A**) global and European scales and (**B**) at an individual scale for the top six European producers. (**C**) Application of the circular economy model for the revalorization of agri-food waste to recover biomolecules of interest for food industry accomplishing (**D**) the first levels of the hierarchical waste model.

**Figure 2 foods-12-03054-f002:**
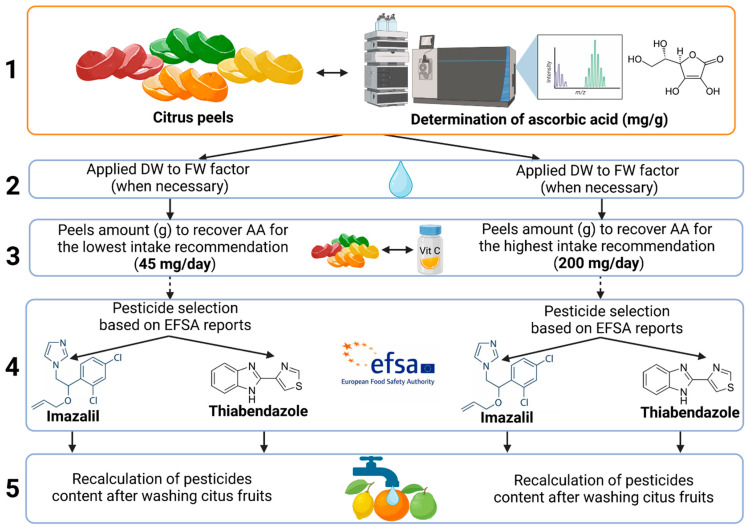
Research stages to perform a safety assessment of orange peels. (**1**) Selection of the by-product and biomolecule of interest. (**2**) Harmonization of ascorbic acid (AA) content by the application of the ‘water content’ factor. (**3**) Determination of the biomass weight required to fulfill the lowest and the highest recommended daily allowances of AA. (**4**) Selection of potential pesticides present in citrus peels: thiabendazole and imazalil. (**5**) Application of processing factor for a better estimation of hypothetical pesticide content.

**Figure 3 foods-12-03054-f003:**
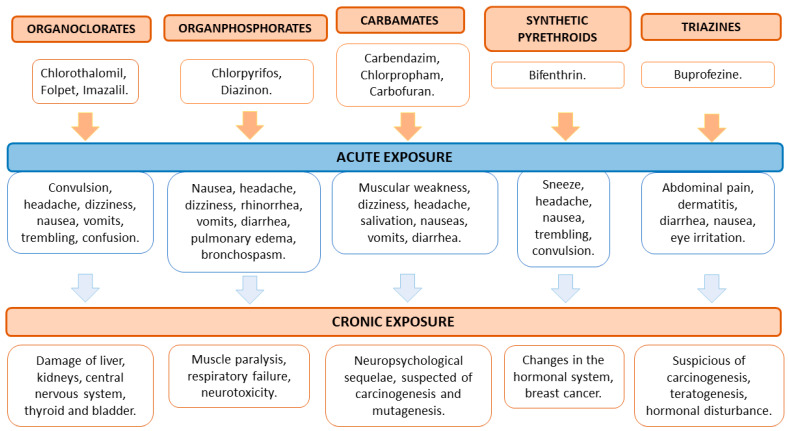
Overview of main chemical groups of pesticides used in Spain, including examples, and their most relevant adverse effects for human health.

**Table 1 foods-12-03054-t001:** Quantitative data related to the production and generation of residues both at global and Spanish level of the main cultured crops in Spain and their potential applications.

Crops	Production		Residues		By-Products	Application	Refs.
	World	Spain	World	Spain			
Orange	70 mT	3.9 mT	30 mT	1–1.8 mT	Pulp, skin, seed	Bio-refinery, bio-compounds, bio-composites, essential oils, bioethanol	[31,32]
Grapes	279 mhL	44.4 mhL	18 mhL	2–3 mhL	Pomace, lees, sludge, scrape	Bio-refinery, bio-compounds	[33]
Strawberry	0.45 mT	0.36 mT	0.05 mT	0.04 mT	Pulp, skin, seeds	Bio-compounds, fiber, colorants, bioethanol	[34]
Red berries	18 mT	0.45 mT	1.8 mT	0.05 mT			
Peppers	34,000 mT	1082 mT	-	-	Seeds, stalks	Biofuel, bio-compounds, fertilizer	[35]
Olives and olive oil	10 mT	6.3 mT	1.3 mT	0.24 mT	Maintenance wastes, pulp, leaves, watermill	Biomass, fertilizer, bio-compounds, plastics	[36,37]
Cereals	1370 mT	23 mT	±50%	±50%	Stalk, peel, pulp, skin, seeds.	Livestock, paper, construction, fuels, fiber, bio-compounds, colorants	[38,39,40]

Abbreviations: mhL: million hectoliters; mT: million tons.

## Data Availability

The data used to support the findings of this study can be made available by the corresponding author upon request.

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
