# Peer review of "Challenges in the Application of Circular Economy Models to Agricultural By-Products: Pesticides in Spain as a Case Study"

_foods, 2023, doi:10.3390/foods12163054_

Round 1
Reviewer 1 Report
Thank you for the opportunity to review the manuscript entitled “Challenges in the Application of Circular Economy Models to Agricultural By-products: pesticides in Spain as a Case Study” (foods-2548931). The manuscript, which is bases on the concepts of circular economy and bioeconomy, as well as on the targets of reutilization and revalorization of biomass, evaluates the most common pesticides used in eight highly productive crops in Spain, in the light of the environmental and the consumers protection. In addition, the research performs a safety assessment of the potential presence of two pesticides when recovering ascorbic acid from citrus peels.
Abstract. The abstract is clear, but it looks a bit redundant. The authors are invited to reduce the information related to the context and the background of the research (LL. 15-27) and to highlight the purpose of the research, the methods adopted and the main insights/outcomes coming from the research. Further, the authors should identify the audience of the research, and its originality in the academia.
Introduction. The section “Introduction” requires several updated and authoritative references, which at current are missing. Also, the section should be restructured, since at current is confusing and not well focused. First, the authors should adopt a clear and comprehensive structure for the “Introduction”, as follows: (i) theoretical context of the research and main challenges to be addressed; (ii) quantitative and qualitative statistics and facts on the topic to be addressed (in this field, for instance, the amount of biowaste and the related environmental, economic and social challenges); (iii) brief presentation of the literature review on the topic; and (iv) purpose of the research, main methods adopted to address the challenges, audience of the research and originality of the study (in the light of the research gaps identified in literature). Such structure is rather needed to develop and clear and consistent “Introduction”.
In the next lines, I identify some minor and major challenges, which I kindly invite the authors to address.
LL. 36-67 should be more referenced, specifically LL. 36-47. In such lines I would include the statistics and facts related to the generation of biowaste (at the global and the local level) and the related environmental, economic and social challenges/implications. Please, consider the subsequent articles, which can help you address clearly the issue.
Rana, R.L., Bux, C., Lombardi, M. (2023). Trends in scientific literature on the environmental sustainability of the artichoke (Cynara cardunculus L. spp.) supply chain. British Food Journal, 125 6, 2315-2332. https://doi.org/10.1108/BFJ-07-2022-0571
Poponi, S., Ruggieri, A., Pacchera, F. and Arcese, G. (2023).The circular potential of a Bio-District: indicators for waste management. British Food Journal. https://doi.org/10.1108/BFJ-12-2022-1137
In addition, the authors are invited to better declare the European strategies towards the achievement of the SDGs, namely the Green Deal, the Farm to Fork Strategy, the Circular Economy Action Plan, the Common Agricultural Policy, the National Recovery and Resilience Plan, etc. Please, consider including the subsequent articles, which are authoritative and updated to the latest legislative and policy issues.
Calabro, G., Vieri, S. (2023). Limits and potential of organic farming towards a more sustainable European agri-food system. British Food Journal. https://doi.org/10.1108/BFJ-12-2022-1067
Passaro, P., Perchinunno, P. and Rotondo, F. (2023). Statistical analysis of the circular economy for the intervention policies of the NRRP. British Food Journal, https://doi.org/10.1108/BFJ-09-2022-0796
LL. 68-95 are quite confusing and should be better focused on the aim and scope of the research. I understand the importance of providing a snapshot of the agri-food production in Europe, but such information, in the section “Introduction”, are rather too much. On the other side, I invite the authors to include a proper section entitled “Theoretical background and literature review”, in which the authors can provide a comprehensive quantitative analysis (also through tables or graphs) of the European agri-food production, as well as the results of a systematic or semi-systematic literature review on the topic (which is missing).
L. 86. It appears “Error! Reference source not found”. Please, revise.
LL. 88-89. It appears “Error! Reference source not found”. Please, revise.
LL. 83-95 should be valorized (in a separated paragraph), in other to highlight the originality/novelty of the current research and justify the choice of Spain as a case-study. Also LL. 96-104 seem rather tautologic. Could you please better clarify such an assumption, in a more concise and consistent manner?
LL. 105-116. It seems rather unfocussed to deal with non-organic waste, since the research evaluates the agri-food by-products. Or the research considers also the non-organic by-products coming from agri-food activities? If the core of the research is the biowaste, such lines are rather useless and could be deleted. I would better jump to L. 116, which deals with the main strategies to valorize “agricultural waste” to produce, for instance, bioenergy or other bio-compounds and biomaterials, which are something different compared to non-organic materials recovery.
L. 121. It appears “Error! Reference source not found”. Please, revise.
Table 1 is not cited in the text. I would remove it or, as suggested, I would include it in the novel section “Theoretical background and literature review”:
L. 132. It appears “Error! Reference source not found”. Please, revise. It appears several times along the manuscript (e.g., LL. 286, 287, 679, 767, 779, 808, 813, 821, 827). It seems that the research has not been sufficiently reviewed before submission, at least in the field of the reference list.
Soon after the section “Introduction”, it appears quite difficult to understand the scientific method adopted by the researchers. There are several sections and sub-sections, but none of them highlights the research methodology adopted in the current research. I would also appreciate the description of the different results and outcomes of the research, and I would save most of them because are interesting and denotates great efforts to investigate the Spanish agri-food sector. However, the authors are invited to describe the research methodology in a scientific manner, as to let readers comprehend the structure of the methodology and allow replicability of the study.
The section “Introduction” should end with the description of the purpose of the research, as well as with the definition of the research methods. At current, I cannot easily undersand the purpose and the methods.
Further, it should be extensively highlighted the nexus between the “re-utilization of residues” in the light of the circular economy and bioeconomy paradigms (LL. 224-226) and the importance of assessing the pesticides content in the agricultural crops in Spain (LL. 227-649).
In the light of the current structure of the research, it seems rather complex to understand the scope of the research. If the purpose is to investigate the amount of pesticides content in the agricultural crops in Spain, in the light of the reuse of biowaste and in the scope to estimate the environmental and (human and animal) health implications of the valorized biowaste in the market, it should be highlighted and clarified.
Also, the analysis of the case-study (LL.757-837) is unclear and difficult to be contextualized in the manuscript.
The general suggestion is to substantially revise the manuscript and re-structure considering the common structure of a scientific article, namely: (i) Introduction; (ii) Theoretical background and literature review; (iii) Materials and methods; (iv) Results; (v) Discussion; (vi) Conclusions. For this reason, the authors are invited to save the most suitable parts of the research and delete the redundancies and the concepts out of scope.
I would appreciate to review the revised version of the manuscript.
English is sometimes challenging and the entire manuscript should be revised, spell-checked and grammar-checked by a native English speaker. I would avoid redundancies. In addition, the authors are invited to revise the references, which sometimes are missing. Please, provide a clear and consistent revision of the manuscript.
Author Response
Response to reviewers
#Reviewer 1 (Please see the attachment)
Thank you for the opportunity to review the manuscript entitled “Challenges in the Application of Circular Economy Models to Agricultural By-products: pesticides in Spain as a Case Study” (foods-2548931). The manuscript, which is bases on the concepts of circular economy and bioeconomy, as well as on the targets of reutilization and revalorization of biomass, evaluates the most common pesticides used in eight highly productive crops in Spain, in the light of the environmental and the consumers protection. In addition, the research performs a safety assessment of the potential presence of two pesticides when recovering ascorbic acid from citrus peels.
Response: Thank you to the reviewer for the effort of performing such a detailed revision of the manuscript. We understand the revision as a process of improvement of the content and the quality of our work, so we hope we had reached this goal after the huge effort we have done to restructure the manuscript. Our responses and corrections are explained below.
Abstract. The abstract is clear, but it looks a bit redundant. The authors are invited to reduce the information related to the context and the background of the research (LL. 15-27) and to highlight the purpose of the research, the methods adopted and the main insights/outcomes coming from the research. Further, the authors should identify the audience of the research, and its originality in the academia.
Response: The content of the abstract has been summarized, the objective of the study underlined, the methods used briefly explained and the main outcomes exposed. Also, the originality of the work has been highlighted.
Introduction. The section “Introduction” requires several updated and authoritative references, which at current are missing. Also, the section should be restructured, since at current is confusing and not well focused. First, the authors should adopt a clear and comprehensive structure for the “Introduction”, as follows: (i) theoretical context of the research and main challenges to be addressed; (ii) quantitative and qualitative statistics and facts on the topic to be addressed (in this field, for instance, the amount of biowaste and the related environmental, economic and social challenges); (iii) brief presentation of the literature review on the topic; and (iv) purpose of the research, main methods adopted to address the challenges, audience of the research and originality of the study (in the light of the research gaps identified in literature). Such structure is rather needed to develop and clear and consistent “Introduction”.
Response: The introduction of the manuscript has been modified. Now all the relative information to the alternative productive systems is presented together along with the reference to the European strategies (SDGs, Green Deal, Farm to Fork Strategy, Circular Economy Action Plan, and the Common Agricultural Policy). Following the introduction now can be found a new subsection “Theoretical background and literature review” to cover the 4 points suggested by the reviewer in this comment and to try to answer the following questions that we understood were linked to the introduction and theoretical background.
In the next lines, I identify some minor and major challenges, which I kindly invite the authors to address.
LL. 36-67 should be more referenced, specifically LL. 36-47. In such lines I would include the statistics and facts related to the generation of biowaste (at the global and the local level) and the related environmental, economic and social challenges/implications. Please, consider the subsequent articles, which can help you address clearly the issue.
Rana, R.L., Bux, C., Lombardi, M. (2023). Trends in scientific literature on the environmental sustainability of the artichoke (Cynara cardunculus L. spp.) supply chain. British Food Journal, 125 6, 2315-2332. https://doi.org/10.1108/BFJ-07-2022-0571
Poponi, S., Ruggieri, A., Pacchera, F. and Arcese, G. (2023).The circular potential of a Bio-District: indicators for waste management. British Food Journal. https://doi.org/10.1108/BFJ-12-2022-1137
Response: The content of lines 36-67 has been fully rephrased and now they contain their respective references. This part of the introduction has been focused to present the challenges to be addressed regarding the current productive systems paying special attention to the environmental scenario. The social and economic aspects are considered just after the references to the European strategies. Whereas the statistics and facts have been included along the new subsection “Theoretical background and literature review”. Regarding the suggested references we have included both works in the introduction at the bottom of the first paragraph for Poponi’s work (L44-49) and at the bottom of the penultimate paragraph for Rana’s work (L83-84).
In addition, the authors are invited to better declare the European strategies towards the achievement of the SDGs, namely the Green Deal, the Farm to Fork Strategy, the Circular Economy Action Plan, the Common Agricultural Policy, the National Recovery and Resilience Plan, etc. Please, consider including the subsequent articles, which are authoritative and updated to the latest legislative and policy issues.
Calabro, G., Vieri, S. (2023). Limits and potential of organic farming towards a more sustainable European agri-food system. British Food Journal. https://doi.org/10.1108/BFJ-12-2022-1067
Passaro, P., Perchinunno, P. and Rotondo, F. (2023). Statistical analysis of the circular economy for the intervention policies of the NRRP. British Food Journal, https://doi.org/10.1108/BFJ-09-2022-0796
Response: European strategies have been explained in the second paragraph of the introduction. Even though the reviewer kindly suggest couple of authoritative and updated articles to facilitate our work, we have preferred to use institutional websites to verify the present state of these policies due to their continuous updating process.
LL. 68-95 are quite confusing and should be better focused on the aim and scope of the research. I understand the importance of providing a snapshot of the agri-food production in Europe, but such information, in the section “Introduction”, are rather too much. On the other side, I invite the authors to include a proper section entitled “Theoretical background and literature review”, in which the authors can provide a comprehensive quantitative analysis (also through tables or graphs) of the European agri-food production, as well as the results of a systematic or semi-systematic literature review on the topic (which is missing).
Response: As suggested by the reviewer we have created a new subsection “Theoretical background and literature review” where it was included the quantitative and qualitative analysis of the status of the agricultural scenario in Europe with special attention to the Spanish panorama. Besides, it was incorporated the potential of the by-products derived from the major agricultural crops in Spain to recover biomolecules., as well as their potential applications in the food industry.
LL. 83-95 should be valorized (in a separated paragraph), in other to highlight the originality/novelty of the current research and justify the choice of Spain as a case-study. Also LL. 96-104 seem rather tautologic. Could you please better clarify such an assumption, in a more concise and consistent manner?
Response: LL83-95: We agree with the reviewer. The new subsection created has now this information (first paragraph) which has been deeper developed and analyzed.
Response: LL96-104: Again, we agree with the reviewer, sentence was not properly redacted. This information is now properly developed along the second paragraph of the new subsection.
LL. 105-116. It seems rather unfocussed to deal with non-organic waste, since the research evaluates the agri-food by-products. Or the research considers also the non-organic by-products coming from agri-food activities? If the core of the research is the biowaste, such lines are rather useless and could be deleted. I would better jump to L. 116, which deals with the main strategies to valorize “agricultural waste” to produce, for instance, bioenergy or other bio-compounds and biomaterials, which are something different compared to non-organic materials recovery.
Response: Again, we agree with the reviewer. This information was unnecessary for not being related with the scope of the manuscript. Therefore, it has been deleted.
Table 1 is not cited in the text. I would remove it or, as suggested, I would include it in the novel section “Theoretical background and literature review”:
Response: We have moved this table to the new subsection. Now is cited.
L. 86. It appears “Error! Reference source not found”. Please, revise.
LL. 88-89. It appears “Error! Reference source not found”. Please, revise.
L. 121. It appears “Error! Reference source not found”. Please, revise.
L. 132. It appears “Error! Reference source not found”. Please, revise. It appears several times along the manuscript (e.g., LL. 286, 287, 679, 767, 779, 808, 813, 821, 827). It seems that the research has not been sufficiently reviewed before submission, at least in the field of the reference list.
Response: We are very sorry for this inconvenience. The reason why it appears this message is because we have linked the text “table” or “figure” to their respective captions, so reader can easily reach them. However, for some reasons we cannot understand this configuration was lost. We have now included the references to tables and figures using plain text to avoid this issue.
Soon after the section “Introduction”, it appears quite difficult to understand the scientific method adopted by the researchers. There are several sections and sub-sections, but none of them highlights the research methodology adopted in the current research. I would also appreciate the description of the different results and outcomes of the research, and I would save most of them because are interesting and denotates great efforts to investigate the Spanish agri-food sector. However, the authors are invited to describe the research methodology in a scientific manner, as to let readers comprehend the structure of the methodology and allow replicability of the study.
Response: We have created a new structure to explain the scientific methodology followed for the development of each section. We hope now it is clearer the manuscript structure and the results that we have tried to summarize in the table 3.
The section “Introduction” should end with the description of the purpose of the research, as well as with the definition of the research methods. At current, I cannot easily undersand the purpose and the methods.
Response: Both the introduction and the new subsection ends with the purpose of the research.
Further, it should be extensively highlighted the nexus between the “re-utilization of residues” in the light of the circular economy and bioeconomy paradigms (LL. 224-226) and the importance of assessing the pesticides content in the agricultural crops in Spain (LL. 227-649).
Response: We have tried to highlight the nexus the “re-utilization of residues” in the light of the circular economy and bioeconomy paradigms and the importance of assessing the pesticides content in the agricultural crops in Spain in the new subsection along the first two paragraphs and the conclusive sentence of this novel section.
In the light of the current structure of the research, it seems rather complex to understand the scope of the research. If the purpose is to investigate the amount of pesticides content in the agricultural crops in Spain, in the light of the reuse of biowaste and in the scope to estimate the environmental and (human and animal) health implications of the valorized biowaste in the market, it should be highlighted and clarified.
Response: At the bottom of the introduction and the new subsection has been incorporated the main purpose of the research.
Also, the analysis of the case-study (LL.757-837) is unclear and difficult to be contextualized in the manuscript.
Response: We have tried to clarify this case-study through the development of the subsection 2.3. in the methods section and the restructuration of the table 4 and the figure 2. Besides we have rephrased the text of the results of the case-study to try to better explain its relevance.
The general suggestion is to substantially revise the manuscript and re-structure considering the common structure of a scientific article, namely: (i) Introduction; (ii) Theoretical background and literature review; (iii) Materials and methods; (iv) Results; (v) Discussion; (vi) Conclusions. For this reason, the authors are invited to save the most suitable parts of the research and delete the redundancies and the concepts out of scope.
Response: We have mainly followed the suggested structure and have revised the full manuscript. The current structure includes: (i) Introduction; (ii) Theoretical background and literature review; (iii) Methods; (iv) Results and discussion; (v) Conclusions.
I would appreciate to review the revised version of the manuscript.
Response: We newly appreciate the effort and the interest on our work.
Comments on the Quality of English Language
English is sometimes challenging and the entire manuscript should be revised, spell-checked and grammar-checked by a native English speaker. I would avoid redundancies. In addition, the authors are invited to revise the references, which sometimes are missing. Please, provide a clear and consistent revision of the manuscript.
Response: The full manuscript has been reviewed including references.

Reviewer 2 Report
Dear Authors,
Following are some comments to improve the paper.
1) Abstract - Does not clarify methodology adopted in the paper. Also, main results and conclusion are missing or are very vague.
2) Introduction section
a) Line 36-45 - You have made few statements without any referencing.
b) The authors need to discuss about waste hierarchy. You have claimed that bio-based model have less carbon footprint, which most of the researchers won't agree. For instance, extracting lycopene from tomato waste can be very carbon intensive and costly process. However, feeding them to black soldier fly, mealworms can result in insect protein for human consumption or as animal feed for poultry, pigs and fishes. Please see paper titled 'Codesign of Food System and Circular Economy Approaches for the Development of Livestock Feeds from Insect Larvae'.
c) There are few 'Error! Reference source not found' throughout your manuscript. Please correct them.
d) Please add structure of the paper.
e) Please discuss what other researchers have done in order to address pesticides, how they tackled it.
3) Section 2 - 'Agri-food by-products as source of natural and healthy bio-molecules' what is the relevance of this section. Why you need it and its importance?
4) Section 3.1 - 3.8 - These are very long. It might be better to present all the info in these section in a table format. It will be easier for readers to follow.
5) Conclusion: This section should be reduced to contain - main highlights of the research and future work and recommendation, if possible.
Kind regards
minor editing of English language is needed.
Author Response
Response to reviewers
#Reviewer 2 (Please see the attachment)
Dear Authors,
Following are some comments to improve the paper.
Response: Thank you to the reviewer for the effort of performing the revision of the manuscript. We understand it as a process to improvement the content and the quality of our work, so we hope we had reached this goal after the huge effort we have done to restructure the manuscript. Our responses and corrections are explained below.
1) Abstract - Does not clarify methodology adopted in the paper. Also, main results and conclusion are missing or are very vague.
Response: The content of the abstract has been summarized, the objective of the study underlined, the methods used briefly explained and the main outcomes exposed. Also, the originality of the work has been highlighted.
2) Introduction section
- a) Line 36-45 - You have made few statements without any referencing.
Response: The introduction of the manuscript has fully been modified. Now all the relative information to the alternative productive systems is presented together along with the reference to the European strategies (SDGs, Green Deal, Farm to Fork Strategy, Circular Economy Action Plan, and the Common Agricultural Policy). Following the introduction now can be found a new subsection (suggested by #Reviewer 1) “Theoretical background and literature review” to cover statistics and facts Therefore, the content of lines 36-45 has been fully rephrased and now they contain their respective references. This part of the introduction has been focused to present the challenges to be addressed regarding the current productive systems paying special attention to the environmental scenario.
- b) The authors need to discuss about waste hierarchy. You have claimed that bio-based model have less carbon footprint, which most of the researchers won't agree. For instance, extracting lycopene from tomato waste can be very carbon intensive and costly process. However, feeding them to black soldier fly, mealworms can result in insect protein for human consumption or as animal feed for poultry, pigs and fishes. Please see paper titled 'Codesign of Food System and Circular Economy Approaches for the Development of Livestock Feeds from Insect Larvae'.
Response: The original text says: “term bio-economy mainly focused on the reduction of GHG emissions and so to counter-act climate change. It covers any sector that uses and produces biological resources to produce food, feed, bio-based products, energy and services. Obtained bio-based products are considered to have a lower carbon dioxide footprint when compared against fossil-based analogues”. Therefore, the claim of “lower carbon footprint” is in reference to any bio-based product derived from the bio-economy. It does not refer to the specific recovery of biomolecules. However, we agree with the reviewer that not all the extractive processes provide greener solutions than alternative uses of this biomass. This statement has been included at the bottom of the section 5 and briefly in the conclusions.
- c) There are few 'Error! Reference source not found' throughout your manuscript. Please correct them.
Response: We are very sorry for this inconvenience. The reason why it appears this message is because we have linked the text “table” or “figure” to their respective captions, so reader can easily reach them. However, for some reasons we cannot understand this configuration was lost. We have now included the references to tables and figures using plain text to avoid this issue.
- d) Please add structure of the paper.
Response: We have mainly followed the suggested structure and have revised the full manuscript. The current structure includes: (i) Introduction; (ii) Theoretical background and literature review; (iii) Methods; (iv) Results and discussion; (v) Conclusions.
- e) Please discuss what other researchers have done in order to address pesticides, how they tackled it.
Response: We have included a discussion regarding the pesticides’ detection performed by other authors. It has been placed at the bottom of the section 5. We appreciate this observation.
3) Section 2 - 'Agri-food by-products as source of natural and healthy bio-molecules' what is the relevance of this section. Why you need it and its importance?
Response: The restructuration of the manuscript has led to the inclusion of this part of the text in the new subsection “Theoretical background and literature review”. It seems important to us to understand what biomolecules are potentially recoverable from each kind of by-product and because it serves as theoretical basis for the safety assessment.
4) Section 3.1 - 3.8 - These are very long. It might be better to present all the info in these section in a table format. It will be easier for readers to follow.
Response: We have tried to summarize this section by including some of these results in the Table 3.
5) Conclusion: This section should be reduced to contain - main highlights of the research and future work and recommendation, if possible.
Response: Conclusions have been rephrased to underline the main highlights and to include some future recommendations.
Kind regards
Comments on the Quality of English Language: minor editing of English language is needed.
Response: The full manuscript has been reviewed including references.

Round 2
Reviewer 1 Report
Thank you for the opportunity to review the revised version of the manuscript entitled “Challenges in the Application of Circular Economy Models to Agricultural By-products: Pesticides in Spain as a Case Study”, submitted for possible publication to Foods. The manuscript has been substantially revised by the authors, who have done several efforts to increase its clarity (above all in shaping the methodological process) and its scientific soundness. The section “Introduction” includes also the theoretical background and the literature review, and the proposal of the case study is clearer and consistent with the purpose of the research. Also, references have been updated with novel and authoritative articles, as well as with official and technical reports. In its current version, I am highly satisfied and I can confidentially suggest its acceptance.
Author Response
We are thankful for the deep revision provided by #Reviewer1 to our manuscript since we sincerely think it has helped to clarify several points and to improve the quality of the content.
Reviewer 2 Report
I am happy with the corrections made to the manuscript as per my earlier comments. However, I find significant amount of typos in your manuscript. This need to be addressed asap.
Significant amount of typos.
Author Response
Reviewer is right, since we used the track changing for highlighting the modifications of the manuscript, many typos were unintentionally created. We have reviewed the full manuscript, we hope we have not missed any.
We want to newly express our thankfulness for the revision provided, we sincerely think it has helped to improve the quality of the manuscript.